# The Accuracy of Lateral Cephalogram for Measuring Alveolar Bone Thickness and Root Diameter on Mandibular Incisors

**DOI:** 10.3390/diagnostics12123159

**Published:** 2022-12-14

**Authors:** Thippawan Limsakul, Pannapat Chanmanee, Chairat Charoemratrote

**Affiliations:** Orthodontic Section, Department of Preventive Dentistry, Faculty of Dentistry, Prince of Songkla University, Hat Yai 90112, Thailand

**Keywords:** accuracy, CBCT, cephalometrics, dentoalveolar position, error in diagnosis

## Abstract

Background: The objective of the study was to ascertain whether the alveolar bone and root of the mandibular central incisor measured from cephalograms can represent the same measurements of both mandibular central and lateral incisors from CBCT. Methods: A total of 38 sets of CBCT images and cephalograms before treatment were selected for this study. Thicknesses included alveolar bone, cortical bone, and cancellous bone at the labial and lingual sides. Root diameter and total root-bone thickness were also evaluated. The measurements were performed at 3, 6, and 9 mm from the cemento-enamel junction. Heights included labial bone height and lingual bone height. All measurements were performed on cephalograms and CBCT images of the mandibular central incisor (L1CT) and mandibular lateral incisor (L2CT). The data were statistically analyzed using one-way ANOVA and Bonferroni tests (*p* < 0.01) to compare the cephalograms, L1CT, and L2CT. Results: The cephalograms presented thicker alveolar bone and cortical bone (labial: 0.16–0.31 mm, lingual: 0.14–0.29 mm; *p* < 0.001) as well as higher alveolar crest (labial: 0.46–0.48 mm, lingual: 0.38–0.39 mm; *p* < 0.001) than the CBCT images on both the labial and lingual sides, whereas lingual cancellous thicknesses were not significantly different (*p* = 0.257). The cephalograms presented greater total root-bone thicknesses than L1CT (0.19–0.30 mm; *p* < 0.001), whereas the cephalograms traced thinner roots than L1CT (0.18–0.23 mm; *p* < 0.001) and L2CT (0.39–0.59 mm; *p* < 0.001). Conclusion: Lateral cephalograms cannot represent both mandibular central and lateral incisor dentoalveolar thicknesses, heights, and root diameters the same as CBCT. However, the differences were less than 0.5 mm.

## 1. Introduction

Alveolar bone is an important factor that indicates the boundaries for orthodontic tooth movement. When the root moves beyond the alveolus and contacts the cortical bone, unwanted orthodontic side effects are likely to occur, such as external root resorption, gingival recession, bone dehiscence, and fenestration [1,2,3].

Cone-beam computed tomography (CBCT) more accurately assesses bony architecture by quantifying bone volume and presenting images in 3D compared to traditional radiographs. Moreover, each tooth can be displayed [4,5,6,7]. However, CBCT delivers higher radiation dose and has a higher cost. Therefore, it must be considered only for critical cases according to the ‘as low as reasonably achievable’ principle to justify, optimize, and limit exposure to radiation.

The lateral cephalogram (Ceph) is a routine radiograph that orthodontists use for diagnosis and treatment planning. However, information from a cephalogram is limited by its two-dimensional nature, magnification, and superimposition of structures. Durao et al. evaluated the accuracy of cephalograms compared to direct measurements of the skull [8]. Measurements from the cephalogram were significantly greater than measurements of the skulls. However, they investigated only the skeletal parameters and did not include the dentoalveolar areas.

Comparisons between cephalograms and CBCT images of the skull and facial bones were conducted and indicated that the values from a cephalogram could be either greater or less than the results from CBCT [9,10,11,12]. The only study that compared conventional cephalograms and CBCT images of dentoalveolar areas was a study by Teerakanok et al. [13], which found that cephalograms presented thicker bone and higher bone height than the CBCT images. However, the study was performed on maxillary incisors.

Mandibular incisor alveolar bone limits labio-lingual incisor movement since it is mostly thin with compact bone. Therefore, tooth movement must be performed cautiously. Furthermore, alveolar bone defects in this area are a common finding before orthodontic treatment, especially on labial surfaces [14,15].

Moving the mandibular incisors either lingually or labially may be indicated in protrusive or retrusive mandibular incisors, respectively. Treatment planning based on a lateral cephalogram uses only the mandibular central incisor that appears on the film. However, in clinical practice, mandibular central incisors and mandibular lateral incisors are moved as one unit. In other words, a mandibular central incisor from a cephalogram has been used to represent both the mandibular central incisor and mandibular lateral incisor.

The root diameter is also important since either a wide or narrow traced root could be associated with either thin or thick adjacent alveolar bone. Therefore, this study measured the alveolar bone thickness and height together with the root diameter from cephalograms and compared the results to the same teeth and mandibular lateral incisors from CBCT images.

## 2. Materials and Methods

### 2.1. Subject Selection

The present study was carried out on cephalograms and CBCT images from pretreatment orthodontic records of patients who attended the orthodontic clinic at the Faculty of Dentistry, Prince of Songkla University from 2014 to 2020. Seventy-one sets of images were in the database of the clinic. The study is a retrospective study that recruited patients from two sources. The first source included subjects who participated in past research [13,16,17]. The second source included patients who required both types of radiographs for treatment planning. The study protocol was approved by the Human Ethics Committee of the Faculty of Dentistry, Prince of Songkla University (EC6305-016).

A total of 38 subjects (13 males, 25 females) with a mean age of 22.13 ± 3.95 years were recruited for this study. The inclusion criteria were (1) healthy adults aged 18–30 years, (2) well-aligned mandibular anterior teeth, (3) no history of facial or dental trauma, (4) no previous orthodontic treatment, (5) no periodontal diseases, (6) no significant medical illness related to bone metabolism, (7) no motion artifact or metal artifact at the mandibular incisors area, and (8) good quality images and contrast resolution. The exclusion criteria were (1) rotation or crowding, (2) history of orthodontic treatment, and (3) previous surgery in the mandibular anterior region.

### 2.2. Lateral Cephalograms

Lateral cephalograms were taken in the natural head position as the reference and obtained using the same orthopantomograph equipped with a cephalostat (GENDEX GXDP-700) at 90 kV, 12.5 mA, 15 s exposure time [18]. The mandibular central incisor and dentoalveolar bone of each cephalogram were traced digitally. All radiographs were digitized and analyzed by Dolphin Imaging^®^ (version 11.9; Dolphin Imaging, Chatsworth, CA, USA). Measurements from the cephalograms were converted to actual distances. A scale ruler on the cephalogram was used to perform the mathematical conversion by ImageJ software version 1.53a (NIH, Bethesda, MD, USA).

The reference line was the long axis of the tooth (Figure 1A). Thicknesses, including total root-bone thickness, alveolar bone, cortical bone, and cancellous bone at the labial and lingual sides, and the root diameter of the cephalometric mandibular central incisor were measured perpendicular to the long axis at 3, 6, and 9 mm apical to the cemento-enamel junction (CEJ) (Figure 1B and Figure 2) [13,19]. Heights including labial bone height and lingual bone height were measured parallel to the tooth axis (Figure 2). All cephalometric parameters were measured in millimeters with two significant digits.

### 2.3. Cone Beam Computed Tomography (CBCT)

CBCT images of the mandibular incisors were scanned using a Veraviewepocs (J Morita Mfg. Corp., Fushimi-ku, Kyoto Japan) at 80 kV, 5 mA, 9.2 s exposure time, 0.125 mm voxel resolution, and 80 × 80 mm field of view. The CBCT data were reconstructed at 0.125 mm increments. Each CBCT scan was oriented along the tooth long axis of the root and the sagittal plane running transversely through the midpoint of the tooth axis (Figure 3A,B). The sagittal plane was used to measure the thickness and height parameters of the left and right central and lateral incisors in each image (Figure 3B,C) following the same vertical references as the lateral cephalograms (Figure 2). All measurements were measured in millimeters with two significant digits by i-Dixel One Volume Viewer software (J Morita Mfg. Corp., Fushimi-ku, Kyoto, Japan). The mandibular central incisor and lateral incisor were assigned the terms L1CT and L2CT, respectively. L1CT and L2CT were average values of the left and right central and lateral incisors.

### 2.4. Statistical Analyses

The Shapiro–Wilk test showed normally distributed variables. Therefore, differences between the three groups (Ceph, L1CT, and L2CT) were tested using one-way analysis of variance (ANOVA) followed by Bonferroni tests. All statistical analyses were performed using SPSS version 17 (SPSS, Chicago, IL, USA). The level of significance of all tests was set at *p* < 0.01.

### 2.5. Sample Size Calculation

The sample size calculation followed a study by Teerakanok et al. [13] to detect a difference between lateral cephalograms and CBCT radiographs with a power of 80%. The calculation indicated that 15 subjects were required.

### 2.6. Quality Control

All measurements were performed by one examiner blinded to the other film of the same subject. To determine measurement error and reliability, 30 randomly selected subjects were remeasured after two weeks. Comparisons between the first and second measurements that used the independent t-test and intraclass correlation coefficient (ICC) illustrated no significant differences between the first and second measurements from the cephalograms as well as the CBCT images. The ICC values between the first and second measurements of the cephalograms and CBCT images were 0.92 and 0.96, respectively, which indicated excellent reliability [20]. No systematic error was observed for any variable in the paired *t*-test (*p* > 0.05). Random errors were estimated by the Dahlberg formula [21], which exhibited measurement errors of 0.12 mm in the cephalometric measurements and 0.05 mm in the CBCT measurements. These random errors were considered acceptable.

## 3. Results

### 3.1. Comparisons between Labial Ceph, L1CT, and L2CT

Comparisons between Ceph, L1CT, and L2CT of the labial side are shown in Table 1. Labial alveolar bone of Ceph, L1CT, and L2CT were only cortical bone at all levels. The labial alveolar bone measurements at the 6 mm level were the narrowest in Ceph, L1CT, and L2CT. The Ceph showed a significantly thicker labial alveolar bone than L1CT and L2CT, but no significant differences were found between L1CT and L2CT at all levels. The differences between Ceph-L1CT and Ceph-L2CT were 0.16–0.19 mm and 0.23–0.31 mm, respectively. The distance from the CEJ to the alveolar crest was the shortest in Ceph, whereas L1CT and L2CT were almost similar. The differences between Ceph-L1CT and Ceph-L2CT were 0.46 and 0.48 mm, respectively.

### 3.2. Comparisons between Lingual Ceph, L1CT, and L2CT

Comparisons between Ceph, L1CT, and L2CT of the lingual side are shown in Table 2. The lingual alveolar bone thickness obviously increased toward the apical area in all groups. Ceph showed significantly thicker lingual alveolar bone than L1CT and L2CT. The significant differences between Ceph-L1CT and Ceph-L2CT were 0.25–0.29 mm and 0.14–0.19 mm, respectively. However, no significant difference was found between L1CT and L2CT. The distance from the CEJ to the lingual alveolar crest was the shortest in Ceph (1.54 mm), whereas L1CT and L2CT were almost similar. The differences between Ceph-L1CT and Ceph-L2CT were 0.38 and 0.39 mm, respectively.

The lingual cortical bone thickness gradually increased toward the apical area in all groups. Ceph showed significantly thicker lingual cortical bone than L1CT and L2CT at all levels. The significant differences between Ceph-L1CT and Ceph-L2CT were 0.23–0.26 mm and 0.14–0.19 mm, respectively. The difference between L1CT and L2CT was not statistically significant.

Lingual cancellous bone did not appear at the 3- and 6-mm levels. Ceph showed significantly thicker lingual cancellous bone than L1CT and L2CT at the 9 mm level. However, no significant differences were found between Ceph-L1, CT Ceph-L2CT, and L1CT-L2CT.

### 3.3. Comparisons of Root Diameters and Total Root-Bone Thicknesses between Ceph, L1CT, and L2CT

Comparisons of root diameters and total root-bone thicknesses between Ceph, L1CT, and L2CT are shown in Table 3. The root diameter gradually decreased from 3 mm apical to the CEJ toward the apical direction in all groups. Ceph was significantly thinner than L1CT and L2CT at all levels. The significant differences between Ceph-L1CT and Ceph-L2CT were 0.18–0.23 mm and 0.39–0.59 mm, respectively. Moreover, L2CT was significantly thicker than L1CT at all levels by 0.21–0.37 mm.

The total root-bone thickness gradually increased from 3 mm apical to the CEJ toward the root apex in all groups. Ceph was significantly thicker than L1CT at all levels (0.19–0.30), whereas Ceph was thinner than L2CT. However, no significant difference was found between Ceph-L2CT. L2CT was significantly thicker than L1CT at all levels. The significant differences between L1CT and L2CT were 0.19–0.37 mm.

## 4. Discussion

Labial alveolar bone measurements at the 6 mm level were the narrowest in Ceph and CBCT images because of the normal architecture of healthy subjects. Previous studies reported bone thickness tended to have a smaller value at the mid-root level than at the coronal third, especially in mandibular lateral incisors with the most prevalent location of fenestration at the mid-root level [22,23]. Labial alveolar bone in Ceph was thicker than L1CT and L2CT at all levels which was similar to a study by Teerakanok et al. [13]. A possible explanation comes from two reasons. First, the thicker Ceph was the result of greater magnification of the cephalogram as reported by Ahlqvist et al. [24] and Rino Neto et al. [25]. Second, the root traced from the cephalogram was significantly narrower than from CBCT, possibly due to errors in tracing the landmarks [26,27]. Furthermore, alignment of the teeth is curved (Figure 4). Therefore, pictures taken from the cephalogram were diagonally skewed, instead of being right in the middle as from CBCT, which caused narrower root diameter. On the other hand, measurements of the alveolar bone thickness included most of the labial and lingual bone boarders, leading to thicker alveolar bone in the cephalogram.

When the labial alveolar height was measured, the alveolar crest of Ceph was incisally higher than L1CT and L2CT, which was in agreement with the findings of Teerakanok et al. [13]. This could be explained by the geometry of the mesial crestal bone of the cephalogram that was higher than the middle crestal bone of the incisor (Figure 5) [28]. The difference of 0.48 mm may or may not be clinically significant depending on the individual periodontium status compared to the healthy distance of the crestal bone to the CEJ which can vary from 1 to 3 mm [29,30,31]. However, the actual labial bone height should be a concern when incisor protraction is planned.

When the labial cortical bone was measured, the thickness had the same explanation as the labial alveolar bone thicknesses because pure cortical bone was found at all measured levels. Cancellous bone was not found on the labial bone at the measured levels because pure cortical bone was found on the labial plate. When pure cortical bone is covered on all labial plates, light force should be used and always monitored [32].

The lingual alveolar bone was thicker toward the apical area in Ceph and CBCT images. Ceph showed a significantly thicker lingual alveolar bone than the CBCT images as well as on the labial side, which affects clinical considerations when planning retraction of the mandibular incisors. The lingual alveolar crest of Ceph was also incisally higher than CBCT images as well as on the labial side, which can be explained by the aforementioned reason. Although a difference of 0.39 mm is not clinically significant, this information concealed the true bone height that orthodontists need to consider carefully when planning a retraction of the mandibular incisor. The lingual cortical bone was gradually thicker in the apical direction in both the Ceph and CBCT images. The thicker lingual cortical bone of Ceph compared to the CBCT images was the result of greater magnification. Lingual cancellous bone presented at the 9 mm level in both the Ceph and CBCT images. The lingual cancellous bone of Ceph was thicker than the CBCT images. However, the differences were very little.

The root diameter gradually decreased toward the apical direction, which was the same as the normal root anatomy [33]. Ceph presented thinner roots than the L1CT. Furthermore, the root diameter of Ceph was thinner than L2CT. Moreover, L2CT was thicker than L1CT because the mandibular lateral incisor presented a thicker root than the central incisor which gave evidence that they are not identical. Previous studies also found that the roots of the mandibular lateral incisors were larger than the mandibular central incisors in the mesiodistal and labiolingual directions [33,34]. In contrast, the roots of the maxillary central incisors were larger than the lateral incisors [13,35].

Total root-bone thicknesses of the Ceph and CBCT images revealed an actual cephalometric magnification, whereas the total root-bone thickness of L2CT was thicker than L1CT.

Even though the mandibular central incisors and mandibular lateral incisors are not identical, the differences in labial/lingual bone thickness and height were very little and not significant. Therefore, it can be reasonably assumed that the bone thicknesses and bone heights of L1CT and L2CT are almost identical. Therefore, they can be used interchangeably in treatment planning. However, orthodontic tooth movement that goes beyond the alveolar bone housing in areas of thin bone may risk causing damage to the root and bone. Even though the differences may be small, the orthodontist should take this into consideration when moving the teeth.

Errors in routine cephalometric tracing can come from either the cephalogram itself due to magnification or distortion [36] and image quality or experience may give rise to errors in landmark identification. Errors may also result from the nature of two-dimensional cephalograms that superimpose other structures, such as adjacent root and periodontal ligament.

The quality of CBCT images depends on many parameters such as tube voltage, tube current, field of view, and voxel size [37]. Voxel size is a key factor determining the spatial resolution of an image [38]. Alveolar bone was evaluated from a previous study which found that 0.4-mm voxel images had obviously lower diagnostic value than 0.125-mm and 0.2-mm images [39,40,41]. In our study, the CBCT voxel size is 0.125-mm, which has enough diagnostic value. All significantly different values compared between Ceph and CBCT in Table 1, Table 2 and Table 3 are larger than the voxel size of 0.125 mm, which implies that the measured differences are accurate.

The limitations of this study should be noted. First, previous studies have not established any norms for comparison, and none of the studied groups can serve as a control group. The real norm could be direct measurements of the skull. A further study using alveolar bone from the skull as the norm is recommended. Second, since the study was conducted within the inclusion criteria, the results are limited to adults with mild anterior crowding and healthy periodontium. The subjects in this study were from the general population and were not classified into different alveolar contours in different facial types or skeletal malocclusions.

## 5. Conclusions

The cephalograms presented thicker bone, higher bone height, and thinner root tracing than CBCT. Bone thickness and height between L1CT and L2CT illustrated no significant differences. The root traced from the cephalogram was narrower than the images from CBCT. The root diameter of L1CT was smaller than L2CT.

## Figures and Tables

**Figure 1 diagnostics-12-03159-f001:**
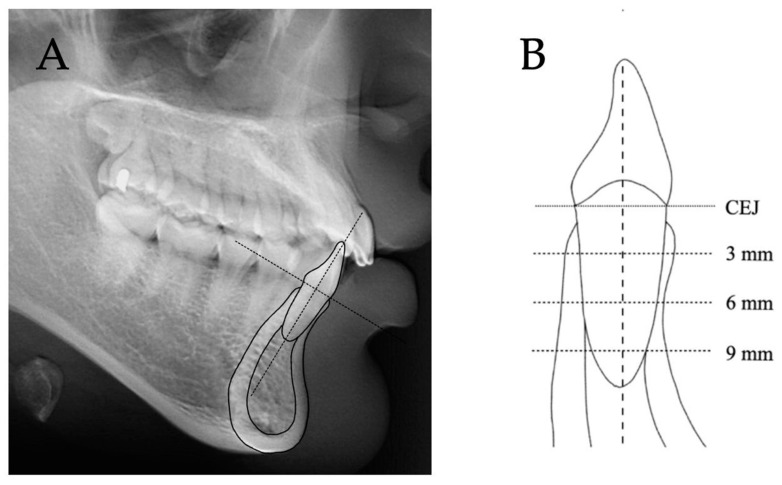
Cephalometric measurements. (**A**) The reference line was the long axis of the tooth. (**B**) Thicknesses were measured perpendicular to the long axis at 3, 6, and 9 mm apical to the CEJ. Heights were measured parallel to the tooth axis.

**Figure 2 diagnostics-12-03159-f002:**
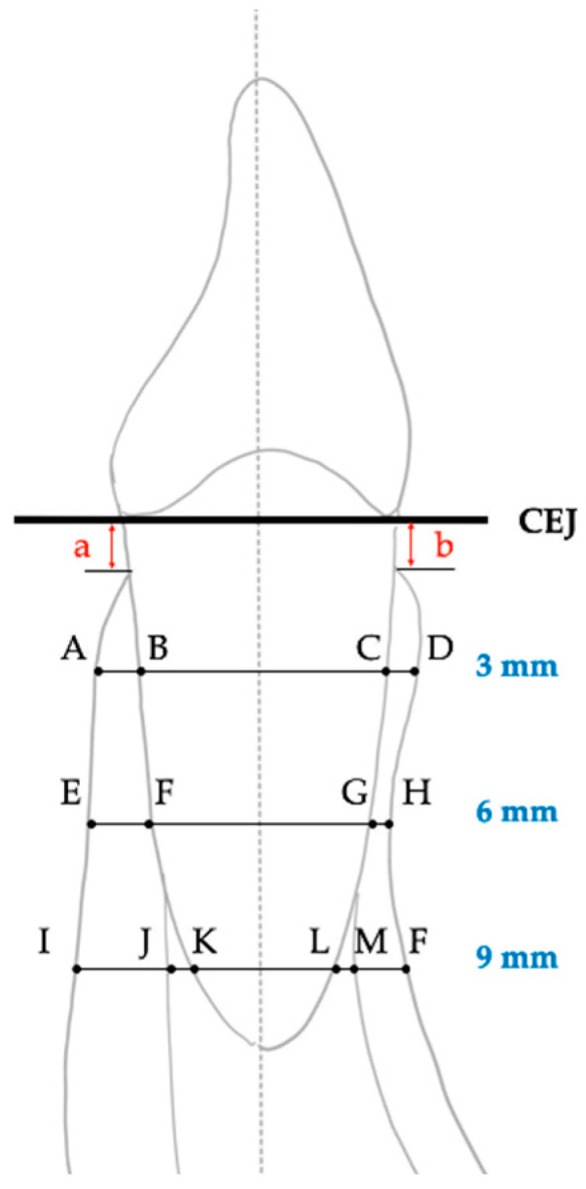
Measurements. On the labial side, labial alveolar bone thicknesses C-D, G-H, and L-F; labial cortical bone thicknesses C-D, G-H, and M-F; labial cancellous bone thickness L-M; labial alveolar bone height distance b. On the lingual side, lingual alveolar bone thicknesses A-B, E-F, and I-K; lingual cortical bone thicknesses A-B, E-F, and I-J; lingual cancellous bone thickness J-K; lingual alveolar bone height distance b; and root diameters B-C, F-G, and K-L and total root-bone thicknesses A-D, E-H, and I-F.

**Figure 3 diagnostics-12-03159-f003:**
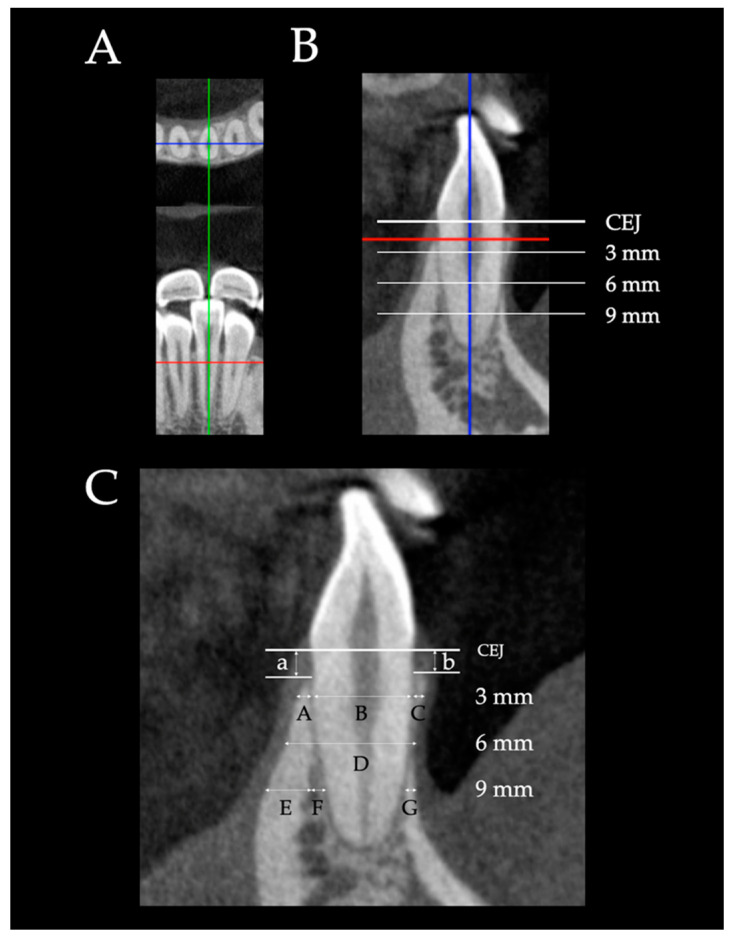
CBCT measurements. (**A**) Tooth orientation of CBCT: The sagittal plane running transversely through the midpoint of the tooth long axis. (**B**) The sagittal plane was used to measure the thickness and height parameters of all four incisors in each image. (**C**) On the labial side, labial alveolar bone thicknesses C and G; labial cortical bone thicknesses C and G; labial alveolar bone height b. On the lingual side, lingual alveolar bone thicknesses A and E+F; lingual cortical bone thicknesses A and E; lingual cancellous bone thickness F; lingual alveolar bone height a; and root diameter B and total root-bone thickness D.

**Figure 4 diagnostics-12-03159-f004:**
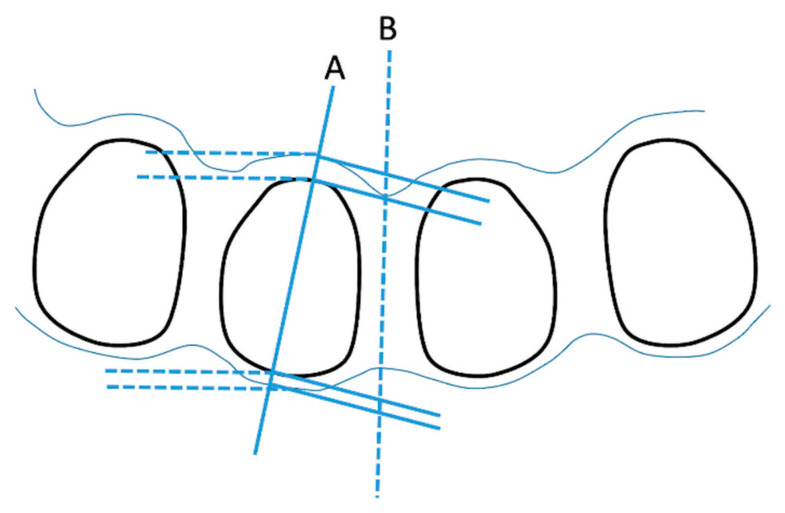
Four mandibular incisor roots were sectioned in the transverse plane to illustrate how the reference plane affects the alveolar bone thickness and root diameter. From the CBCT images, the measurements were performed perpendicular to the cut along the long axis (**A**). From the cephalograms, the measurements were perpendicular to the midline (**B**). The most prominent points in both labial and lingual aspects which appeared on the image were used for the measurements causing thicker bone and narrower root in the cephalograms.

**Figure 5 diagnostics-12-03159-f005:**
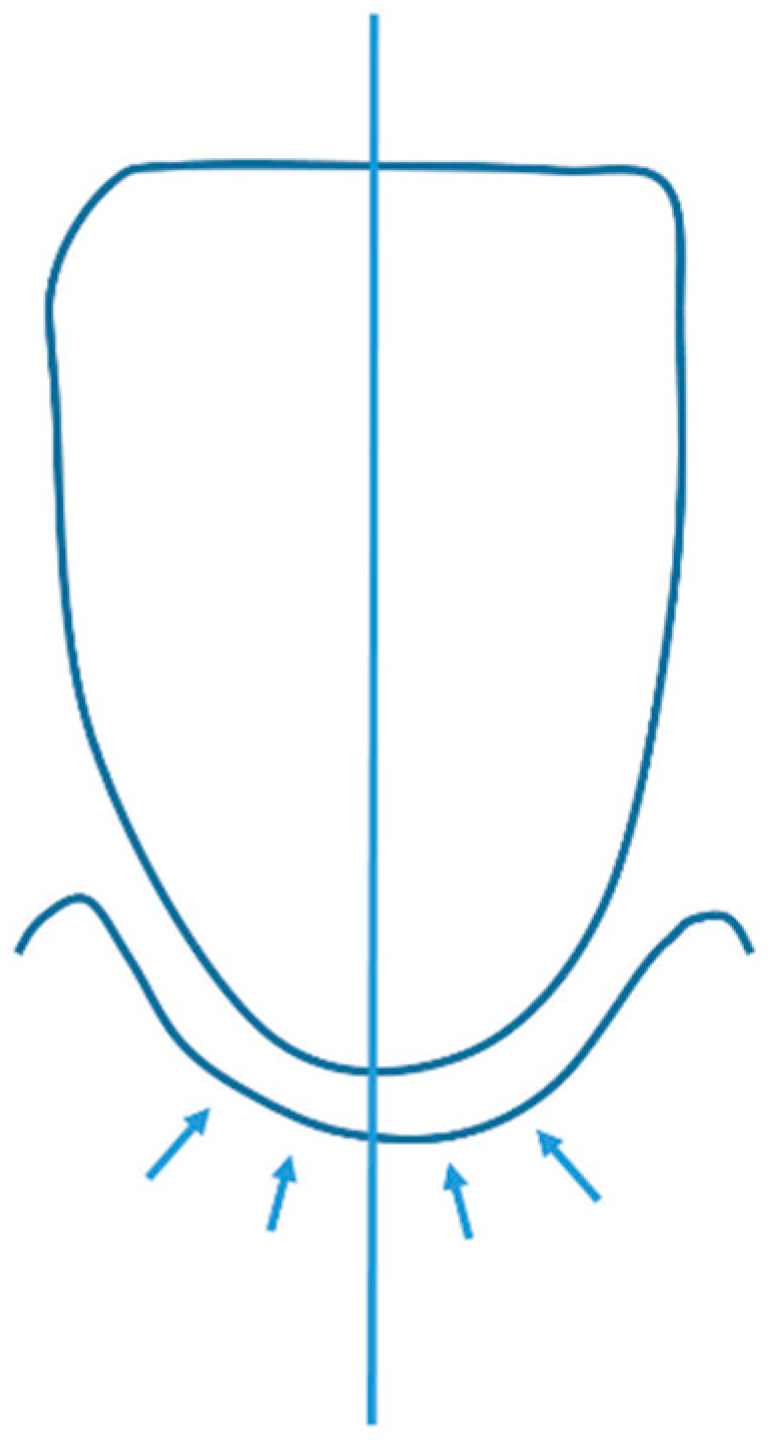
From CBCT. the labial bone height of the mandibular incisor was measured only at the line along the center, whereas from the cephalogram, the occlusal margin of the bone nearby the center of the alveolar crest (arrows) appeared on the film and could be measured.

**Table 1 diagnostics-12-03159-t001:** Comparisons of labial alveolar bone between Ceph, L1CT, and L2CT.

	Labial Side						
Mandibular Teeth (n = 38)	Ceph (C)	L1CT (L1)	L2CT (L2)	ANOVA *p*-Value	Differences		
					C-L1	C-L2	L1-L2
(1) Alveolar bone thicknesses							
3 mm apical to CEJ	0.64 ± 0.08	0.48 ± 0.15	0.41 ± 0.11	<0.001	0.16 ** ± 0.03	0.23 ** ± 0.03	0.07 ± 0.03
6 mm apical to CEJ	0.59 ± 0.18	0.41 ± 0.13	0.30 ± 0.17	<0.001	0.18 ** ± 0.04	0.29 ** ± 0.04	0.10 ± 0.04
9 mm apical to CEJ	1.56 ± 0.13	1.37 ± 0.34	1.25 ± 0.28	<0.001	0.19 * ± 0.05	0.31 ** ± 0.05	0.12 ± 0.05
(2) Alveolar bone height	1.03 ± 0.38	1.49 ± 0.37	1.51 ± 0.39	<0.001	−0.46 ** ± 0.09	−0.48 ** ± 0.09	−0.02 ± 0.09
(3) Cortical bone thicknesses							
3 mm apical to CEJ	0.64 ± 0.08	0.48 ± 0.15	0.41 ± 0.11	<0.001	0.16 ** ± 0.03	0.23 ** ± 0.03	0.07 ± 0.03
6 mm apical to CEJ	0.59 ± 0.18	0.41 ± 0.13	0.30 ± 0.17	<0.001	0.18 ** ± 0.04	0.29 ** ± 0.04	0.10 ± 0.04
9 mm apical to CEJ	1.56 ± 0.13	1.37 ± 0.34	1.25 ± 0.28	<0.001	0.19 * ± 0.05	0.31 ** ± 0.05	0.12 ± 0.05
(4) Cancellous bone thicknesses							
3 mm apical to CEJ	-	-	-	N/A	-	-	-
6 mm apical to CEJ	-	-	-	N/A	-	-	-
9 mm apical to CEJ	-	-	-	N/A	-	-	-

Differences between groups were tested by ANOVA and Bonferroni test. * *p* < 0.01, ** *p* < 0.001.

**Table 2 diagnostics-12-03159-t002:** Comparisons of lingual alveolar bone between Ceph, L1CT, and L2CT.

	Lingual Side						
Mandibular Teeth (n = 38)	Ceph (C)	L1CT (L1)	L2CT (L2)	ANOVA *p*-Value	Differences		
					C-L1	C-L2	L1-L2
(1) Alveolar bone thicknesses							
3 mm apical to CEJ	1.05 ± 0.06	0.79 ± 0.27	0.89 ± 0.04	<0.001	0.26 ** ± 0.04	0.17 ** ± 0.04	−0.10 ± 0.04
6 mm apical to CEJ	1.94 ± 0.24	1.69 ± 0.16	1.80 ± 0.10	<0.001	0.25 ** ± 0.04	0.14 ** ± 0.04	−0.11 ± 0.04
9 mm apical to CEJ	2.83 ± 0.05	2.54 ± 0.25	2.64 ± 0.08	<0.001	0.29 ** ± 0.04	0.19 * ± 0.04	−0.10 ± 0.04
(2) Alveolar bone height	1.54 ± 0.40	1.92 ± 0.39	1.93 ± 0.38	<0.001	−0.38 ** ± 0.09	−0.39 ** ± 0.09	−0.01 ± 0.09
(3) Cortical bone thicknesses							
3 mm apical to CEJ	1.05 ± 0.06	0.79 ± 0.27	0.89 ± 0.04	<0.001	0.26 ** ± 0.04	0.17 ** ± 0.04	−0.10 ± 0.04
6 mm apical to CEJ	1.94 ± 0.24	1.69 ± 0.16	1.80 ± 0.10	<0.001	0.25 ** ± 0.04	0.14 ** ± 0.04	−0.11 ± 0.04
9 mm apical to CEJ	2.14 ± 0.21	1.91 ± 0.35	1.95 ± 0.27	<0.01	0.23 * ± 0.06	0.19 * ± 0.06	−0.04 ± 0.06
(4) Cancellous bone thicknesses							
3 mm apical to CEJ	-	-	-	N/A	-	-	-
6 mm apical to CEJ	-	-	-	N/A	-	-	-
9 mm apical to CEJ	0.69 ± 0.13	0.63 ± 0.16	0.68 ± 0.20	0.257	0.06 ± 0.04	0.01 ± 0.04	−0.05 ± 0.04

Differences between groups were tested by ANOVA and Bonferroni test. * *p* < 0.01, ** *p* < 0.001.

**Table 3 diagnostics-12-03159-t003:** Comparisons of root diameters and total root-bone thicknesses between Ceph, L1CT, and L2CT.

Mandibular Teeth (n = 34)	Ceph (C)	L1CT (L1)	L2CT (L2)	ANOVA *p*-Value	Differences		
					C-L1	C-L2	L1-L2
(1) Root diameters							
3 mm apical to CEJ	5.52 ± 0.15	5.75 ± 0.10	6.00 ± 0.19	<0.001	−0.23 **± 0.03	−0.48 ** ± 0.03	−0.24 ** ± 0.03
6 mm apical to CEJ	4.86 ± 0.09	5.08 ± 0.05	5.45 ± 0.09	<0.001	−0.22 ** ± 0.02	−0.59 ** ± 0.02	−0.37 ** ± 0.02
9 mm apical to CEJ	3.81 ± 0.13	3.99 ± 0.15	4.20 ± 0.17	<0.001	−0.18 ** ± 0.03	−0.39 ** ± 0.03	−0.21 ** ± 0.03
(2) Total root−bone thicknesses							
3 mm apical to CEJ	7.21 ± 0.12	7.02 ± 0.18	7.30 ± 0.15	<0.001	0.19 ** ± 0.04	−0.09 ± 0.04	−0.28 ** ± 0.04
6 mm apical to CEJ	7.39 ± 0.04	7.18 ± 0.37	7.55 ± 0.19	<0.001	0.22 ** ± 0.06	−0.16 ± 0.06	−0.37 ** ± 0.06
9 mm apical to CEJ	8.20 ± 0.12	7.90 ± 0.40	8.09 ± 0.22	<0.001	0.30 ** ± 0.06	0.11 ± 0.06	−0.19 * ± 0.06

Differences between groups were tested by ANOVA and Bonferroni test. * *p* < 0.01, ** *p* < 0.001.

## Data Availability

The data presented in this study are available on request from the corresponding author.

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
