# Peer review of "The Accuracy of Lateral Cephalogram for Measuring Alveolar Bone Thickness and Root Diameter on Mandibular Incisors"

_diagnostics, 2022, doi:10.3390/diagnostics12123159_

Round 1

Reviewer 1 Report

1. the language- some grammar mistakes

2. Figure 1. and 2.

(2) labial alveolar bone height: there was no bone - can you explain

3. Cephalometric measurements on Figure 1 should be traced on lateral cephalograpg (like on CBCT)

4. what was the reason for CBCT -only for research (the study)?

Reviewer 2 Report

The manuscript entitled “The Accuracy of Lateral Cephalogram for Measuring Alveolar Bone Thickness and Root Diameter on Mandibular Incisors” fits into current clinically interesting topics. This study aims to ascertain whether the alveolar bone and the root of the mandibular central incisor measured from cephalograms can represent the same measurements of both mandibular central and lateral incisors from CBCT. The methodology is appropriate and corresponds to recent studies on this topic. However, I have a few questions and comments.

1.      There is no information on the type of ortopantomograph device that was used to record the lateral cephalogram, nor on the applied exposure settings. Specify, as you did for the CBCT.  Please provide the average doses (dose area product)  for patients when recording cephalograms and CBCT scans at the given exposure parameters, so that readers can be aware of the differences in patient dose between the two methods.

2.      In the discussion section (lines 210,211), the authors mentioned the fenestration of labial cortical bone, with the most prevalent location at the mid third of the root. The question is: “Was there any fenestration in your sample, or these patients were excluded from the study? Were those fenestrations, if any, detected using both: cephalogram and CBCT, or only using CBCT? Please comment

3.      In the discussion section (lines 216-218), the authors stated: „Furthermore, alignment of the teeth is curved; Therefore, pictures taken from the cephalogram were diagonally skewed, instead of being right in the middle as from CBCT, which caused narrower measurements of the cephalogram.” I wonder how such an arrangement of the teeth was reflected in the measurement of the other mentioned parameters: alveolar bone height, alveolar bone thickness, and cancellous bone thickness. Please comment, and provide an additional image (as Figure 1.C for example) illustrating the real state of measurement on lateral cephalograms, just as you did for CBCT.

4.      Modify the conclusion so that it repeats the description of the results to a lesser extent. Instead, highlight the clinical relevance of your research and/or write a short recommendation to clinicians.

Reviewer 3 Report

Thank you for giving me the opportunity to evaluate the current submission that compares the accuracy of lower anterior alveolar bone measurement on CBCT vs lateral ceph. Please consider my comments below:

1.     Lines 90-91: the author stated that “measurements from the cephalograms were converted to 100% magnification”. It has been known that lateral ceph image has built-in magnification, and different machines have a different amount of magnifications. In addition, on one lateral ceph image, different anatomic regions also have a different amount of magnifications. please clarify in details how the authors identified the amount of magnification and converted it.

2. figure 1B in the current represents is confusing. As the three horizontal measurement lines are in the same levels as “3mm, 6 mm, and 9 mm” in figure 1A. It looks like measurements 1, 4, 8 were done at the level 3 mm below CEJ; measurement 9 was done at the level 6 mm below CEJ; and measurements 6, 7, 3 were done at the level 9 mm below CEJ. I would suggest considering revising the figure and figure legends.

3.     It’s unclear if the CBCT images for the involved subjects were taken as a routine clinical requirement of pre-orthodontic records, or if they were taken for research purposes.

4.     Line 116-117: please clarify if only the left or right side lateral and central incisors were measured or both sides were measured. If only one side was chosen, what are the decision factors?

5.     For the “differences” in tables 1-3, please state how the numbers were calculated. It should be the average of the differences between two measurements of each sample, then the standard deviation of each difference should also be included in the data presented.

6.     There are some format issues in the tables.

7. In Line 200, there is a typo of “Ceph”.

8.     In the discussion section, the limitation of using CBCT to evaluate the alveolar bone thickness and height should be discussed, as well as the influence of the voxel of the CBCT on alveolar bone measurements.

9.     Overall, the discussion section has a lot of repeated information that is already presented in the result section. Please condense and limited to “discussion”.

10.  Overall, the references are relatively old. Please consider adding more current articles as a significant amount of research on CBCT evaluations in orthodontics has been published in the recent 5 years.

Round 2

Reviewer 3 Report

Thank you for your effort in revising the manuscript. There are some comments that remain and I would suggest the authors consider the following:

1.  Please add the explanation about the 100% magnification in ceph measurements in the revised article.

2. the authors stated that part of the involved subjects are obtained from the previous research project. please cite the article of the previous project, if any.

3. there are typos in table 1

4. it's interesting that Figure 3 clearly shows cancellous bone on both the buccal and lingual sides of the tooth at the level 9 mm below the CEJ, but in Table 1, the authors stated no cancellous bone could be detected in the labial sides

5. with the limitation of CBCT in alveolar bone evaluation, the author should discuss that the differences detected between CBCT and lat ceph measurements are within 1~2-voxel size. Then how accurate are these differences?

Round 3

Reviewer 3 Report

all the comments have been addressed.